# Yoga provision for individuals living with Multiple Sclerosis: Is the future online?

**Gemma Wilson-Menzfeld**[1]*, **Jenni Naisby**[2], **Katherine Baker**[2], **Rosie Morris**[2], **Jonathan Robinson**[3], **Gill Barry**[2]

**1** Department of Nursing, Midwifery and Health, Northumbria University, Newcastle upon Tyne, United Kingdom, **2** Department of Sport, Exercise & Rehabilitation, Northumbria University, Newcastle upon Tyne, United Kingdom, **3** School of Health and Life Sciences, Teesside University, Middlesbrough, United Kingdom

* gemma.wilson-menzfeld@northumbria.ac.uk

## Abstract

### Background

Yoga has multiple benefits for individuals living with Multiple Sclerosis (MS), including reduced pain, depression, fatigue, strength, and improved quality of life. During the COVID-19 pandemic, home-based delivery of yoga increased. However, no studies to date have explored online home-based yoga for individuals living with MS, more specifically the motivations, experiences, or the sustainability of home-based yoga practice for individuals living with MS.

### Aim

This study aimed to explore the facilitators and barriers of online yoga provision for individuals living with MS.

### Methods

One focus group and three semi-structured interviews were carried out online via Zoom with one yoga instructor and seven yoga participants living with MS. Thematic Analysis was used to analyse this data.

### Findings

Two themes were generated from the interviews, the environment and future provision, each with their own sub-themes. The themes reflect various facilitators and barriers of home-based yoga provision which differed depending upon the individuals home environment, social connections, physical ability, and confidence practising yoga. Furthermore, preferences of home provision fluctuated over time depending upon symptoms of MS.

### Conclusions

Home-based yoga practice is a viable and enjoyable option for individuals living with MS. It is recommended that yoga studios offering home-based yoga provision consider individual

**Data Availability Statement:** Relevant excerpts could be made available upon request but only those that have been anonymised. The ethical approval process was dependent upon no raw data

being shared outside of the research team and we have not received ethical approval to store excerpts in a data repository. This study was approved by Northumbria University's ethical approval system. Data requests can be sent to dp.officer@northumbria.ac.uk.

**Funding:** The author(s) received no specific funding for this work.

**Competing interests:** The authors have declared that no competing interests exist.

differences in preference, as well as fluctuations in symptoms that may create inequitable access to services and may prevent participation for some.

## Introduction

Multiple Sclerosis (MS) is an incurable neurodegenerative and chronic inflammatory disease that affects the central nervous system. In Europe and North America, MS is the most common cause of neurological disability in young adults, with diagnosis normally between 20–40 years, and more prevalent in females [1]. Symptoms of MS vary, but can include depression, fatigue, chronic pain, muscle weakness, spasticity, decreased strength, balance and gait problems; all of which led to reduced physical activity and increased risk of falls [2].

Exercise (including balance, strength, aerobic and stretching) is commonly used to improve MS symptoms [3]. Despite evidence for exercise improving MS symptoms [4–6], uptake remains limited. Yoga is a form of exercise that incorporates meditation, breathing exercises, strength and balance, to benefit physical and mental wellbeing simultaneously [7]. Evidence suggests that PwMS find yoga particularly beneficial for reducing pain, depression, fatigue and improving general quality of life and strength [8–10]. A systematic review and meta-analysis of MS and yoga indicated a positive short-term effect on fatigue and mood [11].

Through the COVID-19 pandemic there has been a shift towards home-based consultation and also home-based exercise, especially for more vulnerable groups who were "shielding", including PwMS. Home-based management practices, such as telemedicine, are now being used by People living with MS (PwMS) to manage symptoms, often alongside traditional models of care [12]. Home-based telemedicine allows individuals to partake in remote consultation, assessment, and social networking [12–14], however, there does remain some drawbacks to remote telemedicine practices including low engagement over time [14]. Home-based exercise was also encouraged during lockdown to reducing sedentary behaviour habits [15]. A recent systematic review regarding the effect of home-based exercise for MS found that exercise that is performed between 2–7 times per week can improve health-related outcomes and reduction in fatigue [16]. Furthermore, a Randomised-Controlled trial study looking specifically at home-based yoga for PwMS can have positive outcomes for strength and balance [17]. Home-based exercise combats the barriers of attending face-to-face sessions through accessibility, lack of time, and transportation issues [16, 18]. One disadvantage to home-based exercise is reduced social connections and increased social isolation [18], however, outdoor activities and live home-based exercise programmes, as opposed to pre-recorded classes, went some way to reducing experiences of social isolation [18].

Whilst this research is promising and demonstrates the physical and psychological benefits of home-based practice and the physical benefits of home-based yoga [16, 17], it does not consider individuals' motivations or experiences of home-based yoga practice. Whilst COVID-19 based lockdowns are seemingly coming to an end worldwide, the likelihood of home-based class provision continues. Yet, there is no evidence exploring the motivations, experiences, or the sustainability of home-based yoga practice for PwMS. Therefore, this study aimed to explore the facilitators and barriers of online yoga provision for PwMS.

## Materials and methods

### Design

This study employed a descriptive Phenomenological methodology [19] and a qualitative design. This study was approved by Northumbria University's ethical approval system (Ref: 29257).

## Participants

One instructor (female) and seven class participants took part in this study (5 female; 2 male). All participants were over 18 years old. All class participants self-diagnosed as living with MS and were members of a yoga studio based in London which provided yoga classes targeted at PwMS.

## Data collection

An instructor at the yoga studio acted as a gatekeeper for this study. The instructor sent potential participants an information sheet and consent form via email, explaining the purpose of the study. Participation was entirely voluntary, and potential participants were asked to contact a member of the research team (GB) if they had any questions or wished to participate. The instructor also volunteered to participate in this study. Participants were offered the opportunity to ask questions before providing written consent.

Two members of the research team (GWM; GB), who were independent of the yoga studio, carried out online one-to-one interviews (n = 3) and a focus group (n = 1; 5 participants) using an online video calling platform (Zoom Video Communications, Inc.). The focus group included class participants only. The instructor participated separately in a one-to-one interview. These sessions were set up at the most convenient time for participant, however, due to availability, two other class participants took part in one-to-one interviews rather than the focus group.

The focus group/interview schedule explored motivations to begin yoga, the impact of yoga on MS symptoms, online yoga provision, in-person yoga provision, social connections, and future delivery (Fig 1).

The focus group/interviews lasted between 27–53 minutes. All sessions were recorded using Zoom's recording facility and were downloaded immediately following the session. These recordings were transcribed verbatim.

## Data analysis

Transcripts were uploaded into NVivo 12 (QSR International) for analysis. Braun and Clarke's reflexive, inductive Thematic Analysis was chosen as the data analysis strategy due to its theoretical freedom [20–22]. GWM analysed all transcripts. Initially, the analyst immersed themselves within the transcripts before generating initial codes and subsequent themes [20]. Meetings were conducted with the team to discuss codes and themes before defining and naming themes. Participant quotes were used to demonstrate points of interpretation and generated themes/sub-themes.

## Results

Two themes were generated from the interview data: the environment and future provision. Each theme is made up of multiple sub-themes (Fig 2).

### Theme 1: The environment

The environment played an integral role in online yoga provision. This programme was the first of its kind in the studio, and there were several factors considered for home practice. The participants compared the home environment to the studio environment throughout, and barriers to online provision from home were still present.

**Fig 1. Focus group / semi-structured interview schedule.**

### Setting up for practice at home

The shift from in-person to online yoga provision occurred rapidly due to the onset of the COVID-19 pandemic. This change had not been planned or foreseen and therefore measures were quickly put in place to continue practice despite the implemented restrictions. The yoga instructor visited each participant's home to ensure their space was safe and effective for home practice.

*"I gave them half an hour and we looked in their house [. . .]I helped them" (P001; instructor)*

| The environment | Setting up for practice at home |
| --- | --- |
| | Differentiating practice (home vs studio) |
| | Social interaction |
| Future provision | Live vs recorded online provision |
| | A hybrid approach |

**Fig 2. Themes and sub-themes.**

Some participants owned technology which facilitated their practice, specifically to ensure they were able to be seen by the instructor in all positions. This was one concern for health and safety, as the instructor needed to see individuals in all positions.

*"The main problem is just sort of moving the computer around during the class in order to be seen. So the inversion that I do using my sofa [the instructor] knows I just disappear" (P006)*

Other participants had to invest in technology to sustain their engagement with yoga online. Despite this additional investment, there were no participants who were unable to participate in home practice due to digital exclusion through lack of access or digital skills.

Yoga equipment was another area for investment. Some participants already owned supportive yoga equipment whereas others did not.

*"I had the equipment, but I would stop short like, people are talking about getting ropes and things like that, I'm not, like I'm not, I don't need to get everything"(P002)*

Not everyone's environment was ideal for practising yoga, but the instructor consciously tried to provide alternatives to suit their home environment.

*"So and so goes on the sofa, so and so does it like this [. . .] There are always alternatives [. . .] It doesn't matter so much how they do it, whether it is a pillow or a bolster or this or that, as long as the shape is there, and they get something from it" (P001)*

**Differentiating practice (home vs studio).** It was not just the space that was different from studio to home. The principal difference between the home environment and the studio environment was the absence of physical support, if needed. Support was provided by individuals during in-person classes at the studio. They helped with positioning and with equipment needed during class.

*"When they are in real life class, we have these helpers that, do you need a brick? And they would go and get it. Now they have to get it themselves [. . .] and it works perfectly. So actually, we will learn for when we go back" (P001)*

These changes were not always negative, and learning was taken from home practice to promote independent practice, even when back in the studio environment. However, there were

some downsides in that lack of home support made the instructor, and the participants, wary of more difficult moves, or wary if they felt particularly tired at that time.

*"I will be more cautious and this is generally with all the Zoom teaching, if they say, oh I'd rather not do it today. I may try one more time [. . ..] but if they are then doubtful, I can't do it. Whereas in class I will be next to them saying, do it" (P001)*

This resulted in differing practice from the studio to home. Participants value the physical support provided during in-person classes.

*"My balance is really poor and so I have to adapt what is going on to do the poses etc., how I can do it safely, I think that would probably be easier within the studio around the teachers" (P005)E*

Participants were sometimes more reluctant to do certain poses without the support from instructors. The lack of physical support reduced their confidence of practising some poses, and some described feeling scared when doing more demanding movements.

*"I actually need help to lift the left leg to hold it up, because I can't do it by myself and I'm a bit scared. I won't [go upside down] [. . .] So more confidence, I would say in the studio than by myself here" (P003)*

For others, they did not feel as though they pushed themselves as much when at home, as they did not have instructors observing them.

*"When I'm at home, I'll start doing something like a back bend thing and I'll think, oh, I don't know. Whereas, in class where I'm being observed by three teachers, then you put more effort into it" (P002)*

Although the instructor did encourage this verbally, when appropriate, online.

*"I do have to push them sometimes, but sometimes if they are really too tired [. . .] obviously I am not going to push" (P001)*

All of the participants interviewed in this study had attended the classes for some time and mostly understood their own physical boundaries. The instructor knew them and the physical difficulties they experienced as a result of MS. However, the thought of online provision was proposed as being difficult for new attendees, particularly those living with MS.

*"[The instructor] spends a bit of extra time with [new attendees]. Really focusing on understanding their condition, what they can do [. . .] I think that would be so difficult to do in the Zoom environment" (P005)*

**Social interaction.**    Social interaction is one of the benefits of going to class, especially social interaction with peers who understand each other's condition.

*"Socially because it feels like [. . .] the whole group we're like. . . I accept them as part of my family [. . .] You feel these people, because you see them every day" (P003)*

This social interaction was still valued when online, but it was not seen as being the same as face-to-face interaction.

*"I prefer if we were there to see each other face-to-face. Because you know, you can communicate and chat a little bit or something, but even online it felt, how to say, mentally that you are actually you are not alone, you are doing it with the people you know" (P003)*

The instructor made an effort to incorporate time for social interaction into the online session at the beginning of class, as well as organising additional social sessions, but also recognised that this interaction was dissimilar to in-person classes.

*"Before Christmas we had a little session [. . .] we all introduced ourselves and talked a little bit about who we are and what we are doing and that went through and it was really lovely, but then I felt many people just wanted. . . they were tired, it is also the time of class, they want to eat something" (P001; instructor)*

It was important to consider the social interaction provided by the instructor during the class itself. The "hands-on" support of Iyengar yoga was key for both the instructor and the participants, as well as verbal and visual cues for support. The physical support was not possible, but it was also difficult to communicate between instructor and participants when doing practice online. All participants were encouraged to go on mute so that there was no noise disruption, and the participants could hear instructions, however, this meant that it was not easy to ask questions or advice during class.

*"Because you can't go in and mute yourself that is the other problem with Zoom is that [. . .] it is a big effort to move to the other side of the room to then unmute yourself to say something and then mute yourself again" (P002)*

## Theme 2: Future provision

Participants considered a number of approaches to moving forward, looking at the differences between live and recorded delivery, as well as the potential of offering yoga practice as a 'hybrid' approach of both live online and in-person events.

**Live vs recorded online provision.**   Participants discussed the merits of live online provision, which they considered as being very different to pre-recorded sessions or doing self-led yoga provision. Participants valued the live classes with an instructor and other participants, as opposed to doing yoga by themselves in their own time.

*"This business of having a live person, and [the instructor] is really good at saying, come on stick with it, do it, I can see what you are doing, come on, lift that leg [. . .] For the whole session and [the instructor] keeps you going for the whole lot. And also you can't just get up and clear off" (P007)*

The instructor was also central to online yoga practice for MS due to her understanding of individual barriers. This allowed the instructor to personalise sessions, and poses, for individuals within the class.

*"It is about the amazing opportunity to do these special classes, which with the teacher who understands the conditions that we are dealing with. I wondered how that would work digitally [. . .] I don't know whether it is her magic" (P006)*

Personalisation is not possible during pre-recorded sessions and is a benefit of attending live sessions. Another complication of living with MS can be memory impairment which often made it difficult for participants to do self-led sessions or sessions that are not live.

*"If you are by yourself you just go, eurgh, you just do the first two poses and you can't remember the third one. You are going to have to get up, go to your notes, and look" (P007)*

Despite in-person social interaction being favoured, the social element was important with live classes, and these were therefore preferred to pre-recorded sessions. This online (and in-person) social interaction meant that the participant was sharing the experience with others, and even more meaningfully, with peers living with MS.

*"It is amazing to actually be in a physical space with other people who completely understand the condition [. . .] regardless of whether it's online or in the physical space, to be around people that sort of understand what you're struggling with" (P006)*

**A hybrid approach.** Participants explored the barriers for taking part in both in-person and online provision and there were differing views across participants. There were advantages and disadvantages to both class types. Travel as a major barrier to in-person provision and a benefit to practising from home. Travel could be especially problematic for those living with MS.

*"I will fall over on my way to the tube station. I can't carry my electric scooter up the stairs to the tube station. I will catch cold when I am there and it will take me 10 weeks to get over it" (P007)*

Participant 3 must get a taxi to each class due to restrictions using public transport. Cost is a restriction for this participant, but despite this, they still prefer in-person classes.

*"The travel, this is another thing that is not really convenient for me because I live like 25-minutes by car, half an hour from the institute and it costs me money because I cannot travel by normal transport [. . .] I will do my best to have it face to face. I will make it a priority" (P003)*

Participants suggested the potential of running both online and in-person classes at the same time. There were advantages and disadvantages considered for both modes of delivery.

*"If it is a big class, it wouldn't work so well" (P001; instructor)*

*"You don't get the physical manipulation, but you do get to your class quite easily" (P004)*

*"It increases the access, but the downsides are you don't have that instant, you know, somebody there able to manipulate you or help you or say, we will do something different for you" (P005)*

Whilst challenges to this approach were recognised, the potential to allow for flexibility was also considered.

*"I mean [. . .] obviously for all of us, you know, the more options we have the better" (P006)*

Flexibility of classes was important for PwMS, particularly due to fatigue.

*"You make the effort and try and turn up and there has only been one occasion where I felt literally that I could not do the class. . . and I was really sad about it, you know. So of course, I personally would make as much effort as possible, but it is great if you've got more options, that just goes without saying"* (P002)

Another motivation for a hybrid approach is to have the option of more classes. The instructor also felt that specific classes were more suited to online delivery and would continue to deliver them online.

*"I will continue some kind of class [online]. I do sort of prema yoga and recuperative class and it is beautifully done on Zoom because you don't hear the person next to you snore, you don't smell, you know, you are on your own, you're really on your own. The teacher shows you what to do then they close their eyes and then you just talk through them"* (P001; instructor)

## Discussion

This study aimed to explore the facilitators and barriers of online yoga provision for PwMS. Interviews with participants and the instructor led to the development of two crucial factors: the environment of yoga practice, and future provision of yoga. Various facilitators and barriers were experienced for individuals. Facilitators and barriers varied depending on the individual, but included the home environment, the importance of the instructor, social connections, physical ability, and confidence practising yoga. Furthermore, motivation to practice and ease of practicing yoga at home fluctuated over time through physical and psychological symptoms of MS.

The transition from studio environment to home environment was not always straightforward to begin practising yoga at home. For safety, it was important that the participants could be seen by the instructor. Some participants had to buy both digital and yoga-specific equipment which, once acquired, facilitated home-based practice. It is important that provision is equitable for all, and that individuals who cannot afford extra equipment are not at risk of exclusion from these services and heightened health inequalities. However, a financial benefit related to home-based practice was that for most participants, the cost of home equipment was offset by travel. There were other drawbacks to travel that further increased the preference of online delivery, including accessibility of public transport, particularly when MS symptoms were at their worst. This supports other research examining experiences of homebased exercise for PwMS which documented accessibility, lack of time, and transportation issues as being barriers to exercise classes outside of the home [16, 18]. The yoga instructor was important within online provision, both in terms of setting up the home environment so that it was safe and suitable, and also in their experience and expertise of yoga practice and MS, in understanding participants' strengths and limitations in practice. This allowed the live online yoga sessions to be personalised and suitable to individual class participants.

It was clear that practice changed for some at home, without physical support, and verbal interaction, from instructors or class assistants. This depended upon their confidence, ability to do poses without support, as well as fluctuations of symptoms. Lack of physical support from trustworthy professionals can be a barrier in physical activity engagement for PwMS [23]. A supportive class structure in yoga and sense of belonging have been demonstrated to be key in overcoming barriers to participation for people living with disabilities [24]. The yoga class within this study has been found to provide an online community and the feeling of

support around the individual during a live class, however, for some, this was not comparable to face-to-face sessions.

Whilst there was a sense of community with the online classes, loss of social interaction was an issue with moving from in-person to online yoga classes [18]. Social interaction of live online classes was still valued, and time was set aside to socialise, however it was not always perceived as being good quality social interaction. Community building through meeting up face-to-face and attending social events has been found to enhance adherence to yoga for people living with disabilities [24] and warrants consideration for future models of yoga practice for PwMS.

One consideration for online provision moving forward is digital exclusion. There are many individuals who cannot, or choose not to, use digital technologies, either through lack of access, lack of digital skills, or not recognising the tangible outcomes related to digital technology use [25]. Digital exclusion is associated with inequalities and disadvantages through the life course, and there are multiple factors that increase the likelihood of digital exclusion: educational attainment [26–29], increased age [28–35], health [27, 35–37], disability [30], and gender, with women often being more digitally excluded than men [35–37]. All participants in this study already had broadband access at home and owned at least one digital device which they were able to use to connect to classes online. However, some participants did describe having to invest in technology or yoga equipment to sustain their engagement with online yoga provision. However, for others, the cost of broadband, digital technologies, and yoga equipment, as well as having sufficient space to practise in the home, could lead to exclusion from this future model of provision.

Rather than digitally excluded individuals needing to become digitally connected, services have the responsibility to make services equitable for all, and not to exclude those who do not have the access to digital devices or broadband, or do not have the skills or confidence to utilise digital technology. A 'hybrid' model of yoga practice was suggested in this study, in which both online and face-to-face classes could be ran simultaneously. This has the potential to support participants in both ways, depending on their access needs and preferences. Further research would benefit from evaluating this type of yoga provision for PwMS. Participants valued the flexibility of classes and providing more classes, either in person, online, or both, was valued.

This study has illustrated the benefits and drawbacks of online yoga provision for PwMS. Online yoga provision is a viable alternative to in-person practice, however, the barriers to participation in online yoga must be considered by yoga studios providing these services, to ensure equitability for all class participants, sustainability of practice, and safety. Therefore, it is critical to consider individual preferences and fluctuations in preferences depending upon MS symptoms, as well as the home environment, potential loss of social connection, and exclusion based on lack of digital access or access to equipment.

There are limitations to this study. First, this study used convenience sampling and therefore the sample may not be representative of all PwMS e.g. use of digital technology. Second, this data is collected from one yoga studio, and it is important to consider the perspectives and experiences of PwMS who practise yoga across the UK and internationally. Finally, all of the participants in this study were existing members of the yoga studio and this may have biased the results favourably for at-home practice. Further research must therefore consider the experiences of those who are new to yoga generally, including at-home yoga.

## Conclusions

Our study is the first known study to explore the facilitators and barriers of online yoga provision for PwMS. Data highlighted that home-based yoga practice is a viable and enjoyed option

for PwMS. There were various facilitators and barriers of home-based yoga provision, which were dependent on the individual. Furthermore, preferences for home-based yoga practice were unstable for individuals depending upon physical and psychological symptoms of MS.

For yoga studios offering home-based yoga provision or a hybrid approach of simultaneous in-person and online classes, it is critical to consider these fluctuations, as well as the home environment, potential loss of social connection, and exclusion based on lack of digital access or access to equipment.

## Author Contributions

**Conceptualization:** Gemma Wilson-Menzfeld, Jenni Naisby, Katherine Baker, Rosie Morris, Jonathan Robinson, Gill Barry.

**Data curation:** Gemma Wilson-Menzfeld.

**Formal analysis:** Gemma Wilson-Menzfeld, Jenni Naisby.

**Investigation:** Gill Barry.

**Methodology:** Gemma Wilson-Menzfeld, Gill Barry.

**Project administration:** Gemma Wilson-Menzfeld, Jenni Naisby, Gill Barry.

**Writing – original draft:** Gemma Wilson-Menzfeld, Jenni Naisby, Gill Barry.

**Writing – review & editing:** Gemma Wilson-Menzfeld, Jenni Naisby, Katherine Baker, Rosie Morris, Jonathan Robinson, Gill Barry.

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
