## [Decision Letter · Decision Letter 0]

21 Mar 2022

PONE-D-22-06963Yoga provision for individuals living with Multiple Sclerosis: Is the future online?PLOS ONE

Dear Dr. Wilson-Menzfeld,

Thank you for submitting your manuscript to PLOS ONE. After careful consideration, we feel that it has merit but does not fully meet PLOS ONE’s publication criteria as it currently stands. Therefore, we invite you to submit a revised version of the manuscript that addresses the points raised during the review process.

We look forward to receiving your revised manuscript.

Kind regards,

Luigi Lavorgna

Academic Editor

PLOS ONE

Journal Requirements:

a) Did participants provide their written or verbal informed consent to participate in this study?

Reviewers' comments:

Reviewer's Responses to Questions

**Comments to the Author**

1. Is the manuscript technically sound, and do the data support the conclusions?

Reviewer #1: Yes

2. Has the statistical analysis been performed appropriately and rigorously? 

Reviewer #1: N/A

3. Have the authors made all data underlying the findings in their manuscript fully available?

Reviewer #1: Yes

4. Is the manuscript presented in an intelligible fashion and written in standard English?

Reviewer #1: Yes

5. Review Comments to the Author

Reviewer #1: In my opinion, the fact that the participants were already members of a yoga studio could have influenced positively the results. Discuss it as a limit.

As the authors explored the motivations, experiences, or the sustainability of home-based yoga practice for MS patients, they should mention in the introduction section the role of internet and telemedicine in the management of patients with MS. In this regard, they could add some sentences to introduce this topic (suggested references PMID: 34018047; PMID: 28710056; PMID: 33802029).

“To overcome this barrier, exercise needs to be achievable and effective, but also accepted and engaging to encourage concordance.” This sentence needs a reference

6. PLOS authors have the option to publish the peer review history of their article (what does this mean?). If published, this will include your full peer review and any attached files.

Reviewer #1: No

---

## [Author Response · Author response to Decision Letter 0]

24 Mar 2022

Thank you for taking the time to review this paper. All suggested revisions have been made. They are presented in red text in the manuscript, and a full response to each revision is provided in the accompanying table.

---

## [Editor Report · Decision Letter 1]

28 Mar 2022

Yoga provision for individuals living with Multiple Sclerosis: Is the future online?

PONE-D-22-06963R1

We’re pleased to inform you that your manuscript has been judged scientifically suitable for publication and will be formally accepted for publication once it meets all outstanding technical requirements.

Kind regards,

Luigi Lavorgna

Academic Editor

PLOS ONE
---

## [Editor Report · Acceptance letter]

21 Apr 2022

PONE-D-22-06963R1 

Yoga provision for individuals living with Multiple Sclerosis: Is the future online? 

Dear Dr. Wilson-Menzfeld:

I'm pleased to inform you that your manuscript has been deemed suitable for publication in PLOS ONE. Congratulations! Your manuscript is now with our production department. 

Kind regards, 

on behalf of

Dr. Luigi Lavorgna 

Academic Editor

PLOS ONE